# Quantitative Geochemical Prediction from Spectral Measurements and Its Application to Spatially Dispersed Spectral Data

Andrew Rodger *⬤ and Carsten Laukamp ⬤

Mineral Resources, Commonwealth Scientific and Industrial Research Organisation (CSIRO),
Kensington, WA 6151, Australia; Carsten.Laukamp@csiro.au
* Correspondence: Andrew.Rodger@csiro.au

**Featured Application: Inference of whole rock geochemistry parameters can be inferred from spectral data collected from pulps by combining a dimesionality reduction scheme in the form of non-negative matrix functions and a random forest regression model. The proposed workflow allows the infered geochemical parameters to be used in a smart sampling scheme for both orebody characterisation and/or budgetary constraints. In both cases it can point to areas that do not require intensive sampling while locating areas of mineralisation that are better suited to higher sampling rates.**

**Abstract:** The efficacy of predicting geochemical parameters with a 2-chain workflow using spectral data as the initial input is evaluated. Spectral measurements spanning the approximate 400–25000 nm spectral range are used to train a workflow consisting of a non-negative matrix function (NMF) step, for data reduction, and a random forest regression (RFR) to predict eight geochemical parameters. Approximately 175,000 spectra with their corresponding chemical analysis were available for training, testing and validation purposes. The samples and their spectral and chemical parameters represent 9399 drillcore. Of those, approximately 20,000 spectra and their accompanying analysis were used for training and 5000 for model validation. The remaining pairwise data (150,000 samples) were used for testing of the method. The data are distributed over two large spatial extents (980 km$^2$ and 3025 km$^2$, respectively) and allowed the proposed method to be tested against samples that are spatially distant from the initial training points. Global R$^2$ scores and wt.% RMSE on the 150,000 validation samples are Fe (0.95/3.01), SiO$_2$ (0.96/3.77), Al$_2$O$_3$ (0.92/1.27), TiO (0.68/0.13), CaO (0.89/0.41), MgO (0.87/0.35), K$_2$O (0.65/0.21) and LOI (0.90/1.14), given as Parameter (R$^2$/RMSE), and demonstrate that the proposed method is capable of predicting the eight parameters and is stable enough, in the environment tested, to extend beyond the training sets initial spatial location.

**Keywords:** spectral; geochemistry; random forest; regression; whole rock; MIR; SWIR; VNIR; NMF

## 1. Introduction

The routine collection of spectral reflectance measurements from drillcore and/or laboratory ready samples is now common enough that it is natural to assess the feasibility of using the spectral measurements for quantitative prediction. This action is already performed in various guises with the selected methodology generally based around the desired outcome.

Relatively simple spectral indices have been routinely used within the spectral remote sensing community with great success for many years [1–5] and more recently with proximal spectral sensing of drillcore samples within the exploration community [6–10]. The latter has been driven by the proliferation of hand-held and benchtop spectrometers that have successfully lowered the barrier to entry and provided a means of leveraging

such data for more sophisticated qualitative and quantitative methodologies to aid in the exploration task.

Due to the ease with which spectral data can be collected, and the fast turnaround of higher order products generated from said reflectance data, it can be highly beneficial in providing a means of early confirmation and assessment of a variety of results that may aid in the exploration decision making process [11,12].

While questions pertaining to mineral identification in various mineral systems can be addressed with indices-based methods, the approaches used to relate the bulk, or volumetric, properties of a geological sample to its spectrum require more sophisticated methodologies.

Volume based assessment of geological samples is generally derived from a lab-based analysis of the sample which comes with an inherent cost and turnaround time. While the cost and turnaround time of accessing results from laboratory samples are generally accounted for it does not mean that earlier access to that knowledge would not aid an explorer or decision maker. An earlier preliminary result may, for example, aid in the earlier definition of an existing economic ore body and allow preplanning prior to laboratory based confirmation. Alternatively, it may assist in the process of apportioning which samples are best suited for a deeper laboratory analysis and/or the actual sampling frequency best suited to answering the question at hand.

The use of a partial least squares [13] models for regression analysis relating to geochemical properties is well established [14–17]. These models are generally singular output models and require as many models as the number of parameters that are to be predicted. This use of random forest models [18] have been gaining popularity and applied successfully to both classification [19–21] and regression [22,23] problems. As well as proving to be easily implemented and robust they can also make multioutput predictions and therefore reduce the need for multiple models.

In this study we will investigate a methodology that uses a 2-step process to ascertain if wt.% estimates from whole-rock geochemistry are reliably predictable from spectral measurements of drillcore samples prepared as pulps. The 2-step process makes use of a dimensionality reduction step followed by a multioutput decision tree approach to predict the wt.% of 8 different whole-rock parameters, the majors and LOI, and its predictive effectiveness when applied to a testing set that encompasses a spatial extent extending beyond the initial training and test dataset.

## 2. Materials

The data used in the study are proprietary and are therefore subject to constraints. Namely, the spatial location of the data source cannot be provided without revealing proprietary information.

The data itself represent pulps collected from multiple drillcore which are distributed over 2 large spatial extents of approximately 980 $km^2$ and 3025 $km^2$, respectively. In this study a reference to drillcore sample is given to mean spectral sample as measured from a pulp. The complete dataset is comprised of 7 individual datasets that are made up of spectral measurements and whole rock geochemistry (Fe, $SiO_2$, $Al_2O_3$, $TiO_2$, CaO, MgO $K_2O$ and LOI).

Other variables included in the whole rock geochemistry were P, S and Mn but are not used in this study as they failed to produce a working model. The entire dataset comprises approximately 175 K samples (approximately 25,000 per dataset) with dataset 1 randomly split into 20,000 and 5000 samples for training and validation, respectively. The remaining 6 datasets (approximately 150 K samples) were held out for testing. Table 1 gives the summary statistics comprised of the mean, standard deviation, 50% and 75% quartiles, and the maximum value of the 7 datasets and the 8 geochemical parameters examined in the study.

**Table 1.** The summary statistics comprised of the mean, standard deviation, 50% and 75% quartiles, and the maximum value of the 7 datasets and the 8 geochemical parameters examined in the study.

| Dataset | Statistic | Fe | $SiO_2$ | $Al_2O_3$ | $TiO_2$ | CaO | MgO | $K_2O$ | LOI |
|---------|-----------|------|-------|--------|-------|-------|-------|-------|-------|
| Train/Val | mean | 41.70 | 24.53 | 4.94 | 0.23 | 1.31 | 1.05 | 0.24 | 7.34 |
| | std | 18.67 | 21.91 | 5.98 | 0.39 | 5.17 | 3.46 | 0.73 | 7.44 |
| | 50% | 43.85 | 16.16 | 2.49 | 0.08 | 0.04 | 0.09 | 0.01 | 5.68 |
| | 75% | 58.74 | 44.53 | 6.48 | 0.27 | 0.09 | 0.25 | 0.05 | 8.74 |
| | max | 69.15 | 98.70 | 53.38 | 8.63 | 52.47 | 22.00 | 9.36 | 84.79 |
| Set 2 | mean | 41.18 | 25.83 | 5.32 | 0.27 | 0.98 | 0.80 | 0.17 | 7.21 |
| | std | 17.62 | 21.95 | 6.29 | 0.42 | 4.47 | 3.05 | 0.56 | 7.05 |
| | 50% | 41.74 | 18.93 | 2.71 | 0.09 | 0.04 | 0.08 | 0.01 | 5.63 |
| | 75% | 57.29 | 46.14 | 7.25 | 0.34 | 0.08 | 0.19 | 0.04 | 8.94 |
| | max | 69.29 | 96.51 | 55.92 | 10.10 | 39.70 | 21.50 | 12.10 | 73.82 |
| Set 3 | mean | 43.68 | 22.92 | 4.08 | 0.19 | 0.99 | 0.81 | 0.17 | 7.64 |
| | std | 17.66 | 22.33 | 4.92 | 0.33 | 4.25 | 2.75 | 0.56 | 6.33 |
| | 50% | 47.90 | 11.87 | 2.30 | 0.08 | 0.04 | 0.09 | 0.01 | 6.34 |
| | 75% | 59.18 | 42.94 | 5.14 | 0.20 | 0.09 | 0.22 | 0.04 | 9.21 |
| | max | 68.13 | 96.76 | 51.72 | 7.76 | 49.87 | 20.80 | 11.60 | 73.23 |
| Set 4 | mean | 45.66 | 23.01 | 4.78 | 0.20 | 0.11 | 0.14 | 0.12 | 5.83 |
| | std | 16.12 | 20.52 | 5.70 | 0.31 | 0.96 | 0.57 | 0.44 | 3.05 |
| | 50% | 48.99 | 15.94 | 2.50 | 0.08 | 0.03 | 0.06 | 0.01 | 5.36 |
| | 75% | 59.56 | 40.11 | 6.07 | 0.24 | 0.05 | 0.10 | 0.03 | 7.81 |
| | max | 68.77 | 97.65 | 39.19 | 6.84 | 48.73 | 18.50 | 7.65 | 42.96 |
| Set 5 | mean | 43.36 | 25.37 | 4.57 | 0.22 | 0.34 | 0.32 | 0.13 | 6.44 |
| | std | 15.68 | 21.05 | 5.42 | 0.33 | 2.68 | 1.76 | 0.41 | 4.61 |
| | 50% | 44.33 | 19.45 | 2.36 | 0.08 | 0.01 | 0.05 | 0.01 | 5.64 |
| | 75% | 57.46 | 44.26 | 6.29 | 0.29 | 0.03 | 0.11 | 0.03 | 8.43 |
| | max | 67.32 | 97.73 | 51.14 | 5.96 | 40.53 | 21.00 | 6.85 | 47.01 |
| Set 6 | mean | 43.00 | 23.52 | 5.67 | 0.31 | 0.48 | 0.59 | 0.24 | 6.85 |
| | std | 17.34 | 21.43 | 6.16 | 0.48 | 2.76 | 2.11 | 0.67 | 4.83 |
| | 50% | 45.74 | 14.59 | 3.15 | 0.12 | 0.02 | 0.06 | 0.01 | 6.10 |
| | 75% | 58.11 | 43.35 | 8.75 | 0.46 | 0.06 | 0.21 | 0.10 | 9.09 |
| | max | 69.49 | 96.89 | 51.29 | 25.20 | 42.55 | 20.70 | 6.49 | 46.89 |
| Set 7 | mean | 41.25 | 26.53 | 5.32 | 0.28 | 0.56 | 0.71 | 0.28 | 6.41 |
| | std | 17.46 | 21.85 | 6.14 | 0.45 | 2.96 | 2.29 | 0.74 | 5.33 |
| | 50% | 41.30 | 20.73 | 2.70 | 0.10 | 0.03 | 0.10 | 0.01 | 5.40 |
| | 75% | 56.80 | 46.11 | 7.82 | 0.36 | 0.09 | 0.34 | 0.08 | 8.80 |
| | max | 69.49 | 97.66 | 48.03 | 8.63 | 37.42 | 20.40 | 7.07 | 46.76 |

The spectral data collected from any given pulp sample was via 2 different spectral instruments. The first is the HyLogger [24–26] which collected data in the 350–2500 nm spectral range and whose spectral outputs are given with a 4 nm sampling interval, and the second, a Fourier Transform Interferometer for spectral collection from 2000 nm–25,000 nm with a spectral sampling interval of 3.857 cm$^{-1}$. To create a single spectrum the Fourier Transform Interferometer (FTIR) data from 2000–2500 nm was disregarded, and the remaining spectral signal appended to the HyLogger spectrum.

## 3. Methods

The task is to assess the feasibility of predicting whole rock geochemistry parameters with spectral data used as the driving input to the model/s. The combined HyLogger and FTIR spectral data comprise 1476 spectral bands. To reduce computational overhead and to reduce the dimensionality of the spectra we firstly use a non-negative matrix factorization (NMF) model [27] and follow that with a random forest regression (RFR) model [18] to make our prediction. The model implementations for the NMF and RFR are provided by the python scikit-learn library [28].

### 3.1. NMF

The NMF model is a method of representing data as a linear representation using non-negativity constraints. The imposed non-negative constraint leads to a part-based representation that allows only additive, not subtractive, combinations of the original data [27,29]. Using the NMF in the first step allows us to reduce the dimensionality of the input spectra to a smaller number of components than the entirety of the spectrum and is used as input features to the RFR model.

In the 2-step process we reduce the spectral data to a series of components with the NMF model, and where the components are representations of the additive parts comprising the training signals. Due to the non-negative nature of the components they can have a physically interpretable correspondence [30,31], or when the components are predefined such that the components represent spectral endmembers the parts based weights returned are indicative of the proportions of those endmembers and hence can be used in a linear spectral unmixing [32–36].

The matrix form of the NMF is given by:

$$V = WH \tag{1}$$

where V, of size (# samples, # features), is a linear combination of component weights in W with dimensions of (# samples, # components) and components in H (# components, # features). In this study the number of features in the H matrix are the spectral bands of the spectral signals. In practice the scikit-learn NMF implementation is used in the fitting phase with the spectral data to estimate the W and H matrices of the NMF model. It is the H matrix that we are seeking in this portion of the study.

The 20,000 spectra selected for training were used in the construction of 6 NMF models to compute 6 H matrices where each H matrix differed only in the number of components. The number of components in each H matrix was 5, 10, 15, 20, 25, and 30 components. The 6 individual NMF models were used to transform the 5000 validation spectra and return 6 W component weighting matrices. Each of the 6 W matrices were then inverse transformed to recover the equivalent V matrix which in turn is directly compared to the 5000 validation spectra. We sought to produce a reconstruction score between the measured spectra and those returned by the inverse NMF such that the difference between the measured spectra and the reconstructed spectra are minimized and an $R^2$ score of 0.99 is calculated.

The high valued constraint on the $R^2$ of the reconstruction is set so we are confident that the components in H can represent the measured spectra and provide a reduced dimensionality of spectra fitted to the model via the W matrix. In this study 25 components satisfied the criteria of a 0.99 $R^2$. The NMF model was then established for 25 components and saved so it could be used to transform new unseen spectra.

Although it is not explored further, and as noted earlier, the 25 components in H of the resulting NMF model can be considered spectral end-members of the training set [32–34,37] and the weighting values W returned in a transform the proportion of each endmember required to produce the measured spectrum. In terms of physical size on disk the trained NMF model occupies approximately 300 KB of space.

### 3.2. RFR

With a dimensionality reduction step (NMF model) the resulting 25 weighting values output for a given sample are used as input features to the RFR model to provide prediction values of the whole rock geochemistry. The whole rock geochemistry for the 8 parameters represents the prediction outputs from the RFR model. In a regression scenario random forests, or random decision forests, are an ensemble method that use a collection of decision trees to output the mean prediction of the individual trees [18]. The benefit of using random forests is they are generally considered robust and self-correcting so can reduce the overfitting often observed in individual decision trees [18,38].

While several implementation parameters can be used to construct an RFR model, we have opted to use the defaults of the RandomForestRegressor class in the ensemble module of the scikit-learn library apart from the maximum depth of the decision trees. In the final RFR model the maximum depth of an individual tree was set at 16.

The maximum depth value was determined by increasing the maximum depth of the RFR model until the $R^2$ score of the predictors was found to be minimally different to an RFR model with an unbounded maximum depth on the decision trees. The resulting RFR model was then trained and validated with the 20,000 and 5000 spectral samples, respectively, and saved for future use with the remaining validation data. The physical size of the RFR model on disk is 130 MB.

In summary the application of the 2-step methodology after training and validation is as follows:

1. Set any spectral reflectance values that are less than zero to zero (potential measurement errors). This is a requirement since the NMF cannot work with negative inputs;
2. Transform the N × 1476 (1476 being the total number of spectral bands) individual spectra via the precomputed NMF model to the N × 25 sample space, where N is the number of spectral samples;
3. Input the N × 25 NMF transformed spectra into the trained RFR model as input features and retrieve the estimated parameter values for Fe, $SiO_2$, $Al_2O_3$, TiO, CaO, MgO, $K_2O$ and LOI.

## 4. Results

The results contained herein are split into three subsections. Namely, spectral, global, and downhole. Each subsection focuses on an aspect of the data and/or its relevance to the results. The spectral subsection will look at spectra associated with the eight geochemical parameters and the potential minerals associated with said parameters. The global subsection looks at the performance of the prediction model as it applies to the entire collection of validation data. Lastly, the downhole subsection presents a downhole comparison of predicted results against the measured response of four drillcore.

### 4.1. Spectral

To gauge the potential differences in the spectra associated with a given element, validation dataset four was used to retrieve the spectra corresponding to each of the eight geochemical parameters being at their maximum values. These spectra are shown in Figure 1 where the parameter and value of the maximum for the spectral sample is provided in the legend. To distinguish between the absorptions more easily across the 400–25,000 nm spectral range two separate plots are shown. The upper plot covers the 400–6000 nm and the lower plot the 6000–25,000 nm spectral region. These spectra are not presented to provide an in-depth analysis of the full suite of potential minerals that might be encountered but rather to ascertain if the mineral types are at least consistent with what might be observed when the given geochemical parameter is at a maximum.

Complementary to Figure 1 is Table 2 which provides a summary of some of the major absorption/emission features of minerals present in the study area. It is noted that respective absorption bands can be present in minerals that are not listed in table. The lower and upper wavelength positions are only given for absorption bands where related compositional changes occur otherwise, an estimated central location is provided.

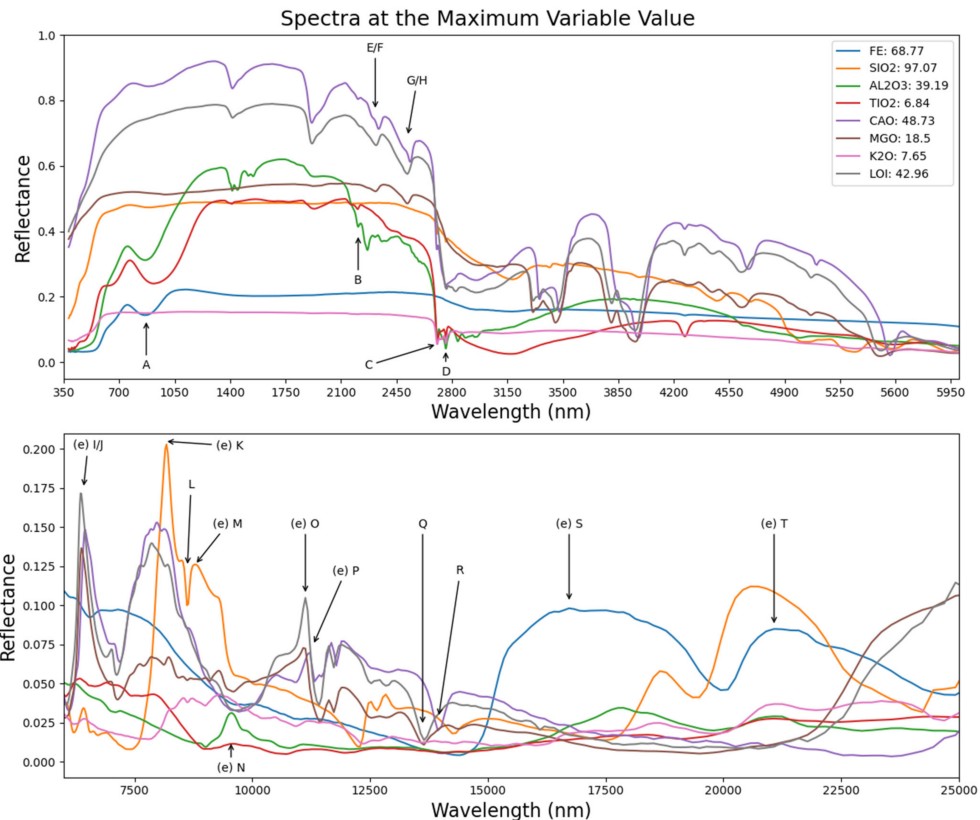

**Figure 1.** Indicative spectra selected from dataset four corresponding to the maximum value of a given geochemical parameters. The spectra are indicators only of the potential differences in reflectance that might be associated with a given parameter. It is expected, and confirmed, that certain whole-rock parameters will be consistent with given mineral assemblages. Complementary to this figure is Table 2 which provides a summary of some of the major absorption/emission features of minerals present in the study area and noted within the spectra.

**Table 2.** A summary of some of the major absorption/emission features of minerals present in the study area. It is noted that respective absorption bands can be present in minerals that are not listed in table. The lower and upper wavelength positions are only given for absorption bands where related compositional changes occur otherwise, an estimated central location is provided.

| Label | Dominant Mineral Group Component/Group | Assignment | Literature | nm/cm$^{-1}$ |
|---|---|---|---|---|
| A | iron oxide/Hematite | $Fe^{3+}$ CFA ($^6A_1 > {}^4T_1$) | [39] | 877/11,403 |
| B | kaolin group/Kaolin | $\nu + \delta Al_2OH_i$ | [40] | 2209/4527 |
| C | kaolin group/Kaolin | $\nu Al2OH_o$ | [40] | 2705/3697 |
| D | kaolin group/Kaolin | $\nu Al2OH_i$ | [40] | 2761/3622 |
| E | Mg-rich Calcium carbonate/Dolomite | $3\nu 3CO_3$ | [41] | 2312–2323/4325–4305 |
| F | Calcium carbonate/Calcite | $3\nu 3CO_3$ | [41] | 2340/4237 |
| G | Mg-rich Calcium carbonate/Dolomite | $2\nu 3 + \nu 1$ | [41] | 2505–2518/3992–3971 |
| H | Calcium carbonate/Calcite | $2\nu 3 + \nu 1$ | [41] | 2530–2541/3953–3935 |
| I | Mg-rich Calcium carbonate/Magnesite | "$\nu 3$ peak" $CO_3$ | [7] | 6405/1561 |
| J | Calcium carbonate/Dolomite | "$\nu 3$ peak" $CO_3$ | [7] | 6490/1541 |
| K | Quartz/Quartz | $\nu$(Si-O-Si) | [42] | 8150/1227 |
| L | quartz/Quartz | $\nu$(Si-O-Si) | [42] | 8598/1163 |
| M | Quartz/Vitreous Silica | $\nu$(Si-O-Si) | [43] | 9025/1108 |

**Table 2.** *Cont.*

| Label | Dominant Mineral Group Component/Group | Assignment | Literature | nm/cm$^{-1}$ |
|---|---|---|---|---|
| N | kaolin group/Kaolin Group | $\nu$Si-O | [44] | 9891/1011 |
| O | Mg-rich Calcium carbonate/Magnesite | "$\nu$2 peak" $CO_3$ | [7] | 11,058/904 |
| P | Calcium carbonate/Dolomite | "$\nu$2 peak" $CO_3$ | [7] | 11,236/890 |
| Q | Mg-rich Calcium carbonate/Ankerite | "$\nu$4 trough" $CO_3$ | [7] | 13,656/732 |
| R | Calcium carbonate/Calcite | "$\nu$4 trough" $CO_3$ | [7] | 13,942/717 |
| S | iron oxide/Hematite | Fe-O lattice vibration | [45] | 16,393/610 |
| T | iron oxide/Hematite | Fe-O lattice vibration | [45] | 22,026/454 |

In Figure 1 the spectra named Fe, $Al_2O_3$ and $TiO_2$ display iron oxide absorptions that are characterized by crystal field interactions around 900 nm [46,47] and are indicative of hematite and goethite. The spectrum associated with the greatest amount of Fe in this case does not appear to have the greatest 900 nm absorption depth, as compared to the $Al_2O_3$ and $TiO_2$ spectra, and is seemingly free from indicative kaolin group absorptions located at 2206 nm [40,47] and 2705 nm [40] and 2761 nm [40] which are present in the $Al_2O_3$ and $TiO_2$ spectra. The Fe spectrum in this case, and because of the lack of other mineral absorptions, is probably a relatively pure Fe sample.

The CaO and MgO (and the sample with the highest LOI) samples are consistent with carbonates. Absorption features associated with carbonates are observed at approximately 2300 nm [41,48], 2500 nm [41], 3500 nm, 4000 nm, 4600 nm and 6400–6500 nm [7]. Calcium dominated carbonates, such as calcite, have absorptions at longer wavelengths in the 2300 nm and 2500 nm spectral regions as opposed to those carbonates, such as siderite or magnesite, where Fe or Mg replaces the Ca, and the absorption features shift to shorter wavelengths [48–50].

As noted, the $Al_2O_3$ spectrum display several absorptions commonly associated with kaolinite but also contain jarosite as defined by a distinct absorption at 2260 nm [51]. The sample associated with the greatest $TiO_2$ has weak kaolin group absorptions at 2160 nm and 2200 nm and around 2700 nm.

The spectrum relating to the highest valued $SiO_2$ is devoid of discernible absorption features in the Visible/Near Infrared/Shortwave Infrared (VNIR/SWIR) but is distinguishable as a quartz sample by the notable peaks located at approximately 8500 nm, 9000 nm, 12,500 nm and 12,800 nm [42,43]. Lastly, the spectrum associated with the highest $K_2O$ value is almost free of any discernible absorption features with the exception being kaolin group absorptions around 2700 nm. In this case the SWIR absorptions around 2200 nm that are also associated with the kaolin group are not discernible.

*4.2. Global*

Table 3 and Figures 2–5 summarise the results of the 2-step workflow, namely NMF-RFR referred to earlier. In Table 3, three separate values are referred to, namely the $R^2$, the root mean squared error (RMSE) and the standard deviation of the RMSE. Column 1 names the dataset in question and lists the number of drillholes that are present in each dataset. Any row that refers to the training dataset are the values as returned by applying the models to the 5000 validation samples while the remaining datasets are the results of applying the models to the unseen testing datasets. The results listed for the "Training" dataset are those values as returned by the validation set (5000 samples) for the RFR model trained on the training set (20,000 samples). All global averages given are the averages for the 6 testing datasets (given as Set2–Set7). In Figures 3 and 5 the results do not include any values from the training/validation dataset and are only comprised of results from testing datasets (Set2–Set7).

An examination of the training/validation data $R^2$ scores in Table 3 and Figure 2 to that of Set2–Set7 shows the $R^2$ is generally maintained in the testing datasets but does

decrease, notably for $TiO_2$ and $K_2O$, compared to the $R^2$ for the training/validation data. This is not wholly unexpected as the spatial locations of the training/validation data are in some cases many tens of kilometers removed from the testing cases.

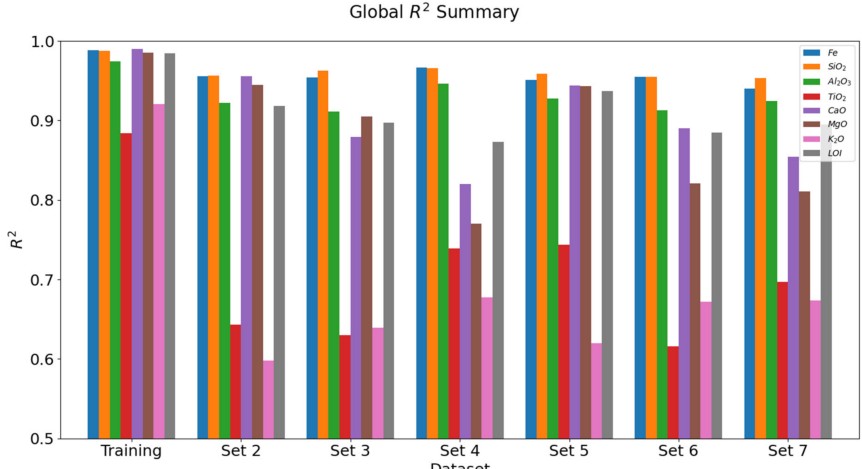

**Figure 2.** The calculated $R^2$ scores for each of the six testing datasets (sets 2–7) and the training and validation dataset (simply given as Training) for the eight whole-rock parameters modeled. The training and validation data return high $R^2$ scores overall which are found to generally decrease when the model was applied to the testing datasets. The reduction in the overall $R^2$ scores was most pronounced in the $TiO_2$, CaO, MgO and $K_2O$ parameters whose values low for most of the data and are considered as consisting of primarily background (refer to Figure 4).

**Table 3.** The $R^2$ score, RMSE and standard deviation of the RMSE for the eight whole-rock geochemical parameters used for training and validation data and the six individual test data sets. The number of drillcore for a given dataset are given in the first column with each dataset comprising approximately 25,000 samples. The results are calculated on a per-dataset basis and represent global results per dataset.

| Dataset | | Fe | SiO₂ | Al₂O₃ | TiO₂ | CaO | MgO | K₂O | LOI |
|---|---|---|---|---|---|---|---|---|---|
| Train/Val: 1185 | $R^2$ | 0.99 | 0.99 | 0.97 | 0.88 | 0.99 | 0.99 | 0.92 | 0.98 |
| Set 2: 1603 | | 0.96 | 0.96 | 0.92 | 0.64 | 0.96 | 0.95 | 0.60 | 0.92 |
| Set 3: 1292 | | 0.95 | 0.96 | 0.91 | 0.63 | 0.88 | 0.91 | 0.64 | 0.90 |
| Set 4: 1087 | | 0.97 | 0.97 | 0.95 | 0.74 | 0.82 | 0.77 | 0.68 | 0.87 |
| Set 5: 1620 | | 0.95 | 0.96 | 0.93 | 0.74 | 0.94 | 0.94 | 0.62 | 0.94 |
| Set 6: 1542 | | 0.95 | 0.96 | 0.91 | 0.62 | 0.89 | 0.82 | 0.67 | 0.88 |
| Set 7: 1070 | | 0.94 | 0.95 | 0.92 | 0.70 | 0.85 | 0.81 | 0.67 | 0.89 |
| | Average | 0.95 | 0.96 | 0.92 | 0.68 | 0.89 | 0.87 | 0.65 | 0.90 |
| Train/Val: 1185 | RMSE | 1.74 | 2.13 | 0.82 | 0.09 | 0.29 | 0.23 | 0.13 | 0.78 |
| Set 2: 1603 | | 3.00 | 3.83 | 1.29 | 0.15 | 0.41 | 0.35 | 0.18 | 1.16 |
| Set 3: 1292 | | 3.21 | 3.84 | 1.25 | 0.13 | 0.74 | 0.55 | 0.25 | 1.42 |
| Set 4: 1087 | | 2.69 | 3.46 | 1.23 | 0.12 | 0.23 | 0.19 | 0.19 | 0.96 |
| Set 5: 1620 | | 2.88 | 3.74 | 1.15 | 0.12 | 0.27 | 0.23 | 0.15 | 0.95 |
| Set 6: 1542 | | 2.91 | 3.71 | 1.30 | 0.13 | 0.32 | 0.32 | 0.18 | 1.09 |
| Set 7: 1070 | | 3.38 | 4.05 | 1.41 | 0.16 | 0.53 | 0.47 | 0.29 | 1.24 |
| | Average | 3.01 | 3.77 | 1.27 | 0.13 | 0.41 | 0.35 | 0.21 | 1.14 |
| Train/Val: 1185 | Sdev RMSE | 0.99 | 1.21 | 0.42 | 0.10 | 0.43 | 0.32 | 0.16 | 0.49 |
| Set 2: 1603 | | 1.62 | 1.95 | 0.86 | 0.15 | 0.71 | 0.53 | 0.27 | 1.20 |
| Set 3: 1292 | | 2.01 | 1.99 | 0.71 | 0.14 | 1.36 | 0.69 | 0.27 | 1.39 |
| Set 4: 1087 | | 1.26 | 1.65 | 0.55 | 0.09 | 0.40 | 0.22 | 0.19 | 0.54 |
| Set 5: 1620 | | 1.68 | 1.86 | 0.68 | 0.11 | 0.51 | 0.31 | 0.16 | 0.56 |
| Set 6: 1542 | | 1.39 | 1.72 | 0.82 | 0.14 | 0.61 | 0.56 | 0.23 | 0.76 |
| Set 7: 1070 | | 1.93 | 1.95 | 0.84 | 0.16 | 0.78 | 0.66 | 0.29 | 0.92 |
| | Average | 1.65 | 1.85 | 0.74 | 0.13 | 0.73 | 0.49 | 0.24 | 0.89 |

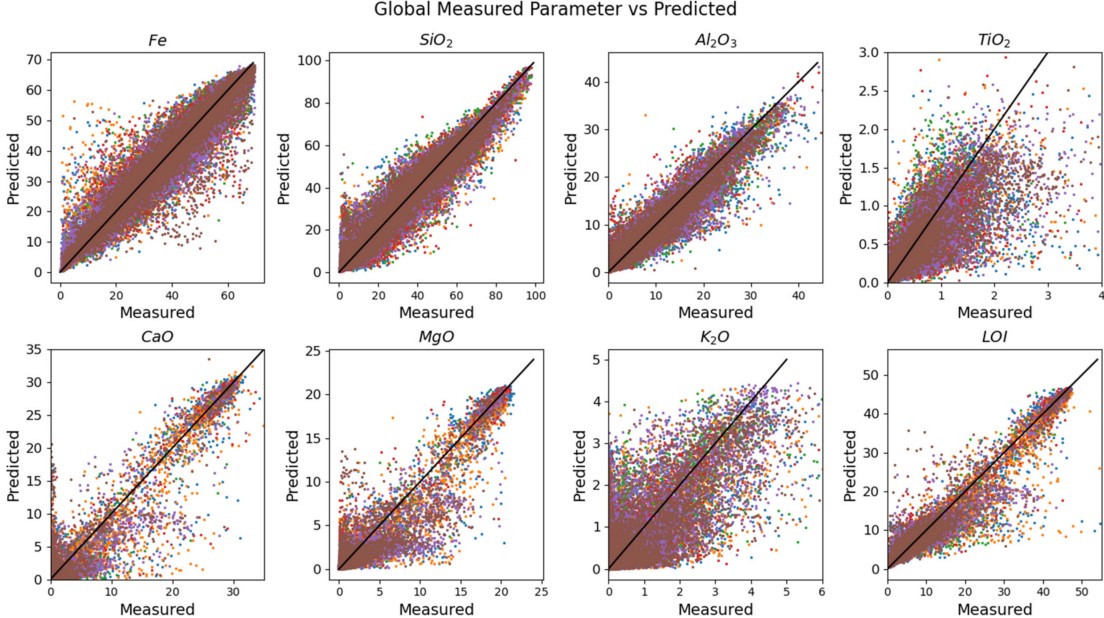

**Figure 3.** Measured whole-rock parameters versus the predicted value for the eight geochemical parameters used (wt.%). Each plot shows the combined measured versus predicted for the six valuation datasets only. The 1-to-1 line is shown in black for each plot. While CaO and MgO show what appear to be appreciable ranges the bimodal nature of the values is also observed with the bulk of the being primarily distributed around the origin i.e., background values (refer to Figure 4).

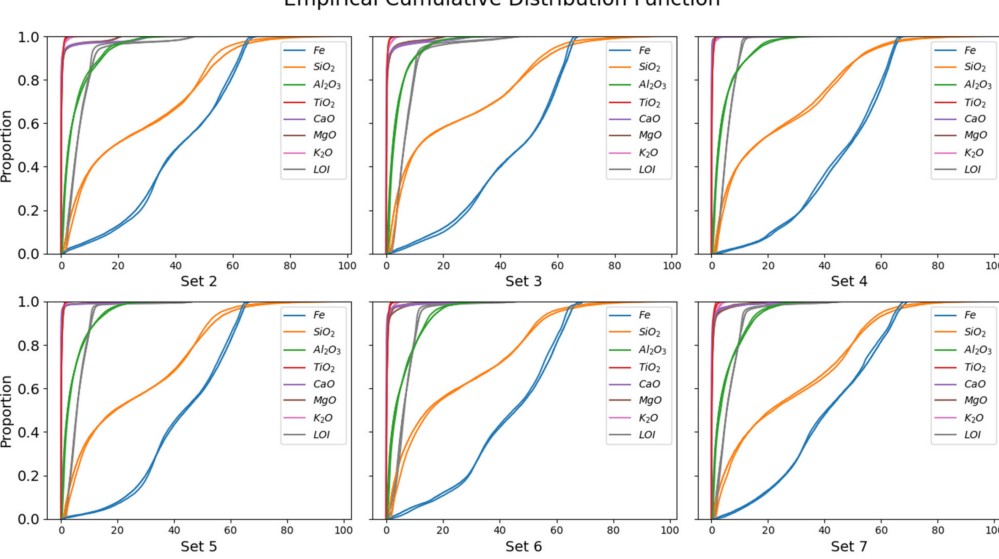

**Figure 4.** The empirical cumulative distribution of the 8 whole-rock geochemical parameters for the 6 validation datasets (sets 2–7) and the predicted values as returned by the proposed method. A successful prediction over a given dataset should produce a distribution that is the same as the actual distribution. Generally small departures are noted and would indicate that the model is accurately reproducing the true distribution and values. It is noted that the distributions for $TiO_2$, CaO, MgO and $K_2O$ show the range of values for these parameters is extremely small and primarily confined to background.

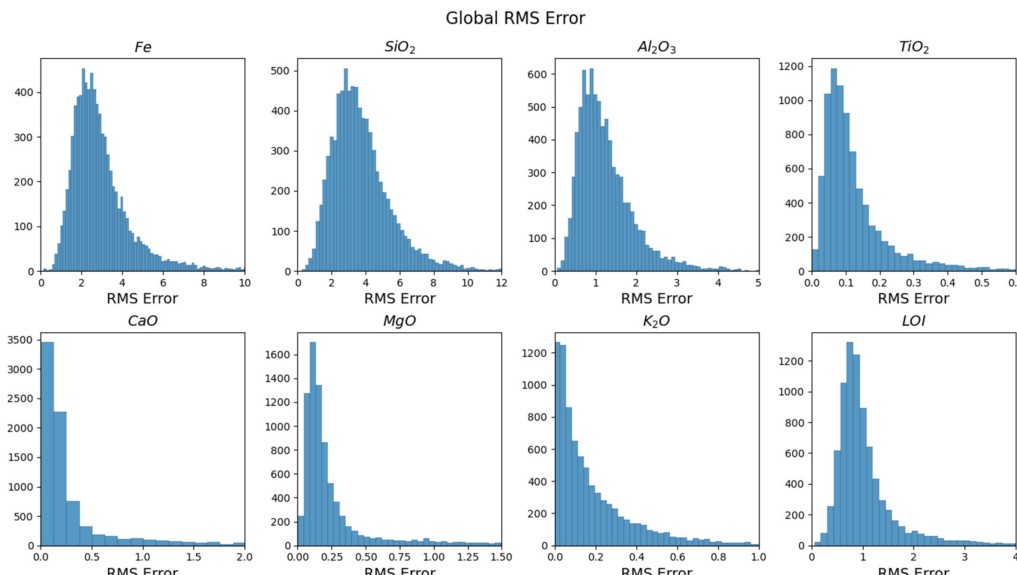

**Figure 5.** The RMSE (wt.%) distribution of the eight whole-rock geochemistry parameters as calculated from the 6 validation datasets. In this case the RMSE is that reported on a per-drillhole basis so the spread of potential RMSE can be evaluated. While the $R^2$ scores (Figure 1) for $TiO_2$, CaO, MgO and $K_2O$ were found to generally be smaller than the other four majors the small RMSE and known distribution still indicate a successful modeling.

Figure 3 shows the measured versus predicted values for the 6 validation data sets with the black line in each plot representing the 1-to-1 line. Specifically, the range of values for $TiO_2$ and $K_2O$ are seen to be small as compared to the other parameters with the bulk of the $TiO_2$ and $K_2O$ heavily clustered near the origin. The lack of defining range for these two parameters would seem a likely contributing factor to their decreased $R^2$ values. CaO, MgO and the LOI have a level of bimodal distribution (see Figure 3) and while they also are heavily distributed near the origin the bimodality most likely helps to extend the range and provide clearer paths for the decision trees within the RFR. The bimodal distribution of the LOI, which corresponds with the MgO and CaO distributions, aligns with the spectral examples given in Figure 1 where the spectrum from dataset four that corresponding to the greatest LOI value is a carbonate dominated spectrum such as the MgO and CaO spectra.

Figure 4 shows the estimated cumulative distribution function for the 6 validation datasets and the eight geochemical parameters. This plot (estimated from kernel density estimators) allows a comparison of the predicted value distributions to the measured. A successful prediction for a given dataset should show the same distribution without major deviations. In general, the distributions follow each other for a given dataset and parameter indicating that the combined NMF-RFR model is working reasonably well.

Figure 5 presents the distribution of RMSE for each of the predicted parameters over the six validation datasets. No distinction in this figure is made between the six datasets and the results are therefore global. In most cases the central RMSE is small compared to the overall range of values for a given parameter. However, and as noted previously, the clustering of values for $TiO_2$ and $K_2O$ around the origin implies the RMSE for these parameters is relatively larger than their counterparts.

### 4.3. Downhole

Lastly, shown in Figure 6 are the downhole predicted and measured values for Fe, $SiO_2$, $Al_2O_3$ and LOI of 4 drillcore. The drillcore shown are not from any one dataset and were selected based on their length (randomly selected from all drillcore that had greater than 150 entries) to show the ability of the model. The other four geochemical parameters are not shown since the scale of the plots reduces those traces to lines just above zero.

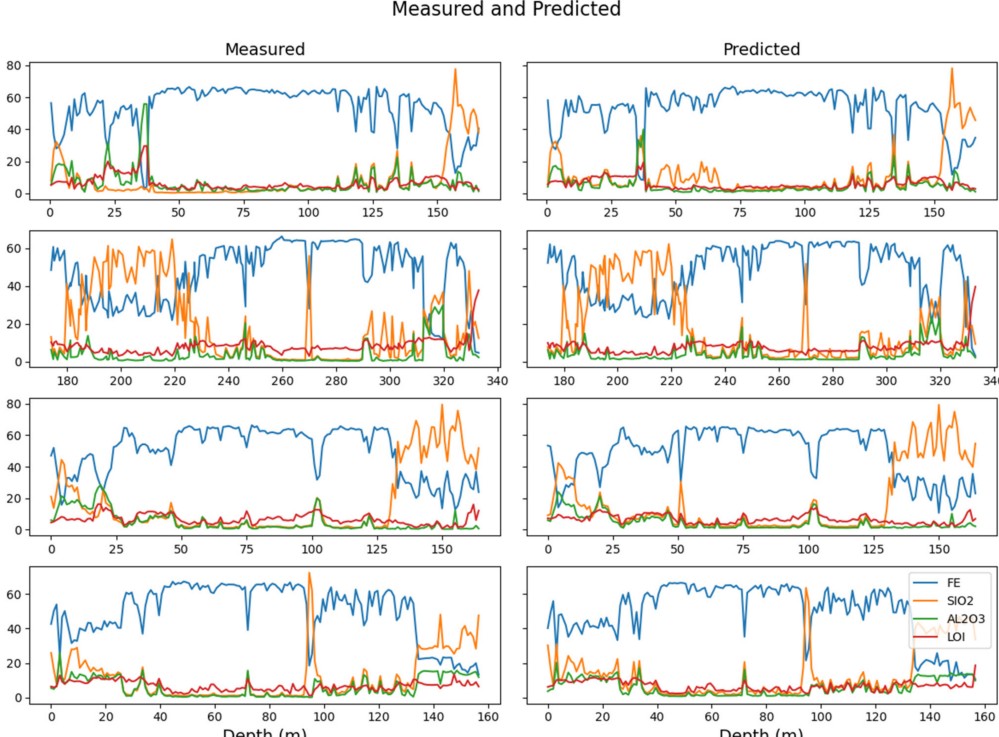

**Figure 6.** The measured and predicted downhole values for Fe, SiO$_2$, Al$_2$O$_3$ and LOI of four randomly selected drillcore where the length of the drillcore comprised greater than 150 entries. The other whole-rock geochemistry parameters are not shown due to the values being comprised of background only which only appear as singular valued traces along y = 0. Overall a good correspondence between the predicted and measured values is observed with generally minor deviations noted.

The measured values are shown on the left-hand side and the predicted values on the right. The y-axis on all measured-predicted pairs is the same for ease of comparison. The performance of the model is generally observed to be good and matches the measured values well. Some discrepancies can be found but overall, the geochemical parameters as predicted from the spectral input could most certainly be used to ascertain the downhole distribution and value of said parameters.

## 5. Discussion

The aim of this study was to ascertain if geochemical parameters could be predicted from spectral measurements. The spectral measurements in this study have a 1-to-1 correspondence with whole-rock geochemistry and provide a data rich avenue for investigating the feasibility.

The global findings demonstrated that a relatively small amount, compared to the total number of samples in the entire dataset, of spectral samples and matched geochemistry can be used to successfully train a combined NMF and RFR model to make accurate and quantitative predictions. The dataset used for this study are from a reasonably uniform and non-diverse geology and allowed accurate predictions to be made that extended far beyond the spatial confines of the training and test dataset locations. If the underlying geology was to markedly depart from that of the training and test data used to build the models, then it would require new models to be built that can incorporate such changes.

However, not all the whole-rock entries were modeled well or even able to be modeled. To successfully predict quantitative values for a given element requires that the element in question has a broad range. By this it is meant that if an element, such as P or S, are not well represented within the geology and samples then there is, as intuitively expected, nothing to model. In this study several geochemical elements had a small range of values

and/or where distributed close to the origin. These represented background values of which larger values were not always present, as was the case with $TiO_2$ and $K_2O$, and for which the predictive power of the model with these elements was limited. Other elements such as CaO and MgO demonstrated bimodal distributions whose range, even though the bulk of the values are scattered accumulated about the origin, allowed a reasonable quantitative prediction to be made. In most cases the RMSE on those elements which were range restricted were still small and can be used confirm that the element is of background quantities.

This aspect was reiterated by producing per-drillcore $R^2$ scores and RMSE values. Those elements that had an extensive range, such as Fe and $SiO_2$, produced per-hole high $R^2$ scores and small RMSE values. On the contrary, and specifically for those elements that were range constrained, poor $R^2$ scores (while still producing low RMSE in most cases) result. However, when drillcore are encountered with an extended range the individual $R^2$ increases and the RMSE also slightly increase.

While this may seem an obvious result it is worth noting and bearing in mind that if one trains and validates a model and applies it, for example to a single drillcore, the returned result, if it was compared later to measured values for that same single drillcore, may appear to be poor. In other words, producing an $R^2$ score on the singular drillcore should not be used as an indicator of model performance.

Additionally, the use of a RFR model in this case has proven to be successful but it has limitations that may necessitate the retraining of the model at future dates. While the RFR is robust it does not extrapolate and hence cannot return predictions beyond the largest and smallest values used to train the model. Thus, if the initial data used to train the model is a subset of a greater range, then the model would need retraining to account for the extended range. Indicators this may be needed are results being returned that are consistently at the extent of training data's range.

In this study the spectral data cover a comprehensive wavelength range that might not be considered typical. However, the principle applied should be viable for reduced spectral ranges such as those encountered by the HyLogger only or by FTIR only. It is expected that a reduced range, and hence a lack of absorptions features that are representative of various elements, may lead to a reduction in the overall accuracy of the model depending on the element sought and the spectral range considered. For example, attempting to quantify $SiO_2$ from the VNIR/SWIR spectral range may prove to be extremely difficult due to the lack of absorption features associated with $SiO_2$ in that spectral range. Future work will test this hypothesis by constraining the data to reduced ranges to evaluate the impact on the regressions.

## 6. Conclusions

This work represents a new and novel approach to the prediction of whole rock geochemistry from spectral measurements. While previous works have used NMF as a method of spectral unmixing they have not as far as we are aware utilized the weights W as inputs to a RFR, to predict whole rock geochemistry.

In summary a viable method of reliably predicting several whole rock geochemistry parameters from spectral measurements of pulps has been defined and validated against a much larger spatially distributed dataset. Of the eight parameters modeled, four show exceptional promise and have validation $R^2$ scores greater than 0.8 and RMSE in the low single digit range. Of the other four parameters the $R^2$ were less but the RMSE scores possibly still acceptable. The proposed method could be used to return a quick turnaround of potential downhole distributions and might be used to better spend an analysis budget. Namely, by highlighting spatial regions prior to laboratory based whole rock analysis more focus, through the laboratory analysis, can be made of those areas deemed to be of economic importance. Conversely, areas identified by the proposed method of having no economic importance might be subject to laboratory analysis at a reduced sampling space. The use of the NMF model to reduce the dimensionality of the spectral measurements was also shown

to be effective and provided reductions in dimensionality, and hence the number of input features to the RFR, by a factor of approximately 59. This has the effect of reducing the computational burden and reducing the overall size of the models. Additionally, the use of preexisting software libraries means that such workflows are accessible to everyone and easily implemented.

**Author Contributions:** Conceptualization, A.R.; methodology, A.R.; software, A.R.; validation, A.R. and C.L.; formal analysis, A.R. and C.L.; writing—original draft preparation, A.R.; writing—review and editing, A.R. and C.L. All authors have read and agreed to the published version of the manuscript.

**Funding:** This research received no external funding.

**Data Availability Statement:** The data used in this study are subject to proprietary constraints and are therefore not publicly available.

**Conflicts of Interest:** The authors declare no conflict of interest.

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
