# Peer review of "Quantitative Geochemical Prediction from Spectral Measurements and Its Application to Spatially Dispersed Spectral Data"

_applsci, doi:10.3390/app12010282_

Round 1

Reviewer 1 Report

Formulate an algorithm for processing large amounts of data for users.

Author Response

done. The paper represents an algorithm for the processing of large data volumes for the prediction of whole rock geochemistry.

Reviewer 2 Report

Line 77: Superscript 2 (error occurs twice on that line)

Line 80: subscript the appropriate chemical formula numbers

Line 245: Superscript the 2

Line 369: Subscript the 2

Author Response

Hi,

Thank you for your review. Much appreciated.

All comments and suggestions made by this reviewer have been implemented. They were also picked up by a second reviewer as well.

Regards

Andrew

Reviewer 3 Report

In this manuscript, the aim of this study was to ascertain if geochemical parameters could be predicted from spectral measurements. The spectral measurements in this study have a 1-to-1 correspondence with whole-rock geochemistry and provide a data rich avenue for investigating the feasibility. The efficacy of predicting geochemical parameters with a 2-chain workflow using spectral data as the initial input is evaluated. Spectral measurements spanning the approximate 400-25000nm spectral range are used to train a workflow consisting of a non-negative matrix function (NMF) step, for data reduction, and a random forest regression (RFR) to predict 8 geochemical parameters. The manuscript offers an interesting topic with interesting conclusions. Before considered publication, the manuscript needs to be improved into a publishable form. Please make sure that your manuscript fully proves that this work is fundamentally novel. Specific comparisons should be made with previously published materials that have similar purposes. Explain how significant progress has been made in this work. Please ensure that your summary and conclusion not only summarize the main findings of your work, but also explain how this work fundamentally advances the field compared with previous literature.

Some more specific critiques:

  1. Line 77: “km2” should be “km2”.
  2. Line 80-81: “SiO2, Al2O3, TiO2 and K2O” should be “SiO2, Al2O3, TiO2 and K2O”.
  3. Line 108: Please add the matrix expression and algorithm of the NMF model, and explain the calculation steps of the parameters and components selected in this study using the NMF model in line 121 to 132.
  4. Line 130: “R2” should be “R2”.
  5. Line 140: Please add the random forest regressor algorithm, and explain the calculation steps of the parameters and components selected in this study using the RFR model in line 141 to 157.
  6. Line 318: Please separate the discussion and conclusions into two sections. In addition, a description is added to the discussion content to make a specific comparison with the previously published materials that have similar purposes, and explain the significant progress that has been made in this work.
  7. Line 319-361: Different minerals have different reflection spectral characteristics. Through spectral analysis, processing the scanning spectrum of rock, analyzing various information such as mineral composition and grade, and then determining the distribution of different minerals at different depths according to the core location, it can be realized the qualitative and quantitative research on the physical and chemical conditions and temporal and spatial changes of altered ore.
  8. Line 369: “SiO2” should be “SiO2”.
  9. Interference spectrum detection is realized by calculating the proportion of core coverage area in the spectral scanning field of view. When the area proportion is lower than a certain threshold, the scanning spectrum is determined as the interference spectrum. This method needs to solve two core problems: how to extract the core coverage area and how to calculate the core coverage area. Whether the interference spectrum is discussed in this study, and the discussion is added to explain the relationship between the interference spectrum and this study.

Author Response

Dear reviewer,

thank you so much for your input. We have made the majority of your suggestions. Specifically we have included (in the discussion) why this is a new and novel method and advances the field. To the specific critiques:

1->4: all suggestion and critiques implemented.

5: The algorithm, specifically the library used from sklearn, was added. The actual hyperparameters etc were also discussed in the text.

6: Separated the discussion and conclusions into 2 sections. Great suggestion.

7: Taken as comment as we think that is what is being offered here.

8: Done

9: Taken as comment as we think that is what is being offered here.

Round 2

Reviewer 3 Report

Dear authors, thank you for addressing my comments.